# DepWiGNN: A Depth-wise Graph Neural Network for Multi-hop Spatial Reasoning in Text *

**Shuaiyi Li[1], Yang Deng[2,†], Wai Lam[1]**

[1] The Chinese University of Hong Kong, [2] National University of Singapore

li.shuaiyi@link.cuhk.edu.hk, ydeng@nus.edu.sg, wlam@se.cuhk.edu.hk

## Abstract

Spatial reasoning in text plays a crucial role in various real-world applications. Existing approaches for spatial reasoning typically infer spatial relations from pure text, which overlook the gap between natural language and symbolic structures. Graph neural networks (GNNs) have showcased exceptional proficiency in inducing and aggregating symbolic structures. However, classical GNNs face challenges in handling multi-hop spatial reasoning due to the over-smoothing issue, *i.e.*, the performance decreases substantially as the number of graph layers increases. To cope with these challenges, we propose a novel **Dep**th-**Wi**se **G**raph **N**eural **N**etwork (**DepWiGNN**). Specifically, we design a novel node memory scheme and aggregate the information over the depth dimension instead of the breadth dimension of the graph, which empowers the ability to collect long dependencies without stacking multiple layers. Experimental results on two challenging multi-hop spatial reasoning datasets show that DepWiGNN outperforms existing spatial reasoning methods. The comparisons with the other three GNNs further demonstrate its superiority in capturing long dependency in the graph.

## 1 Introduction

Spatial reasoning in text is crucial and indispensable in many areas, *e.g.*, medical domain (Datta et al., 2020; Massa et al., 2015), navigations (Zhang et al., 2021; Zhang and Kordjamshidi, 2022; Chen et al., 2019) and robotics (Luo et al., 2023; Venkatesh et al., 2021). It has been demonstrated to be a challenging problem for both modern pretrained language models (PLMs) (Mirzaee et al., 2021; Deng et al., 2023a) and large language models (LLMs) like ChatGPT (Bang et al., 2023). However, early textual spatial reasoning datasets, *e.g.*,

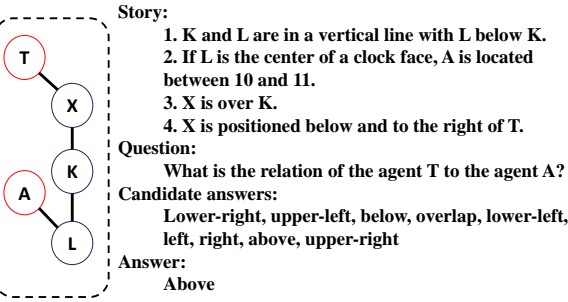

**Story:**
  1. K and L are in a vertical line with L below K.
  2. If L is the center of a clock face, A is located between 10 and 11.
  3. X is over K.
  4. X is positioned below and to the right of T.
**Question:**
  What is the relation of the agent T to the agent A?
**Candidate answers:**
  Lower-right, upper-left, below, overlap, lower-left, left, right, above, upper-right
**Answer:**
  Above

Figure 1: An example of multihop spatial reasoning in text from the StepGame dataset (Shi et al., 2022).

bAbI (Weston et al., 2016), suffer from the issue of over-simplicity and, therefore, is not qualified for revealing real textual spatial reasoning scenario. Recently, researchers propose several new benchmarks (Shi et al., 2022; Mirzaee and Kordjamshidi, 2022; Mirzaee et al., 2021) with an increased level of complexity, which involve more required reasoning steps, enhanced variety of spatial relation expression, and more. As shown in Figure 1, 4 steps of reasoning are required to answer the question, and spatial relation descriptions and categories are diverse.

To tackle the problem of multi-hop spatial reasoning, Shi et al. (2022) propose a recurrent memory network based on Tensor Product Representation (TPR) (Schlag and Schmidhuber, 2018a), which mimics the step-by-step reasoning by iteratively updating and removing episodes from memory. Specifically, TPR encodes symbolic knowledge hidden in natural language into distributed vector space to be used for deductive reasoning. Despite the effectiveness of applying TPR memory, the performance of this model is overwhelmed by modern PLMs (Mirzaee and Kordjamshidi, 2022). Moreover, these works typically overlook the gap between natural language and symbolic relations.

Graph Neural Neural Networks (GNNs) have been considerably used in multi-hop reasoning (Xu et al., 2021b; Chen et al., 2020b; Qiu et al., 2019).

*This work is substantially supported by a grant from the Research Grant Council of the Hong Kong Special Administrative Region, China (Project Code: 14200719)

† Corresponding author.

These methods often treat a single graph convolutional layer of node information propagation (node to its immediate neighbors) in GNN as one step of reasoning and expand it to multi-hop reasoning by stacking multiple layers. However, increasing the number of graph convolutional layers in deep neural structures can have a detrimental effect on model performance (Li et al., 2018). This phenomenon, known as the over-smoothing problem, occurs because each layer of graph convolutions causes adjacent nodes to become more similar to each other. This paradox poses a challenge for multi-hop reasoning: *although multiple layers are needed to capture multi-hop dependencies, implementing them can fail to capture these dependencies due to the over-smoothing problem.* Furthermore, many chain-finding problems, *e.g.*, multi-hop spatial reasoning, only require specific depth path information to solve a single question and do not demand full breadth aggregation for all neighbors (Figure 1). Nevertheless, existing methods (Palm et al., 2018; Xu et al., 2021b; Chen et al., 2020b; Deng et al., 2022) for solving this kind of problem usually lie in the propagation conducted by iterations of breadth aggregation, which brings superfluous and irrelevant information that may distract the model from the key information.

In light of these challenges, we propose a novel graph-based method, named **Dep**th-**Wi**se **G**raph **N**eural **N**etwork (DepWiGNN), which operates over the depth instead of the breadth dimension of the graph to tackle the multi-hop spatial reasoning problem. It introduces a novel node memory implementation that only stores depth path information between nodes by applying the TPR technique. Specifically, it first initializes the node memory by filling the atomic information (spatial relation) between each pair of directly connected nodes, and then collects the relation between two indirectly connected nodes via depth-wisely retrieving and aggregating all atomic spatial relations reserved in the memory of each node in the path. The collected long-dependency information is further stored in the source node memory in the path and can be retrieved freely if the target node is given. Unlike typical existing GNNs (Morris et al., 2019; Velickovic et al., 2017; Hamilton et al., 2017; Kipf and Welling, 2017), DepWiGNN does not need to be stacked to gain long relationships between two distant nodes and, hence, is immune to the over-smoothing problem. Moreover, instead of

aimlessly performing breadth aggregation on all immediate neighbors, it selectively prioritizes the key path information.

Experiments on two challenging multi-hop spatial reasoning datasets show that DepWiGNN not only outperforms existing spatial reasoning methods, but also enhances the spatial reasoning capability of PLMs. The comparisons with three GNNs verify that DepWiGNN surpasses classical graph convolutional layers in capturing long dependencies by a noticeable margin without harming the performance of short dependency collection.

Overall, our contributions are threefold:

- We propose a novel graph-based method, DepWiGNN, to perform propagation over the depth dimension of a graph, which can capture long dependency without the need of stacking layers and avoid the issue of over-smoothing.

- We implement a novel node memory scheme, which takes advantage of TPR mechanism, enabling convenient memory updating and retrieval operations through simple arithmetic operations instead of neural layers.

- DepWiGNN excels in multi-hop spatial reasoning tasks, surpassing existing methods in experimental evaluations on two challenging datasets. Besides, comparisons with three other GNNs highlight its superior ability to capture long dependencies within the graph. Our code will be released via https://github.com/Syon-Li/DepWiGNN.

## 2 Related works

**Spatial Reasoning In Text** has experienced a thriving development in recent years, supported by several benchmark datasets. Weston et al. (2016) proposes the bAbI project, which contains 20 QA tasks, including one focusing on textual spatial reasoning. However, some issues exist in bAbI, such as data leakage, overly short reasoning steps, and monotony of spatial relation categories and descriptions, which makes it fails to reflect the intricacy of spatial reasoning in natural language (Shi et al., 2022). Targeting these shortages, StepGame (Shi et al., 2022) expands the spatial relation types, the diversity of relation descriptions, and the required reasoning steps. Besides, SPARTQA (Mirzaee et al., 2021) augments the number of question types from one to four: *find relation* (FR), *find blocks* (FB), *choose objects* (CO), and *yes/no* (YN), while

SPARTUN (Mirzaee and Kordjamshidi, 2022) only has two question types (FR, YN) built with broader coverage of spatial relation types.

**Tensor Product Representation** (TPR) (Schlag and Schmidhuber, 2018a) is a mechanism to encode symbolic knowledge into a vector space, which can be applied to various natural language reasoning tasks (Huang et al., 2018; Chen et al., 2020a). For example, Schlag and Schmidhuber (2018a) perform reasoning by deconstructing the language into combinatorial symbolic representations and binding them using Third-order TPR, which can be further combined with RNN to improve the model capability of making sequential inference (Schlag et al., 2021). Shi et al. (2022) used a paragraph-level, TPR memory-augmented way to implement complex multi-hop spatial reasoning. However, existing methods typically apply TPR to pure text, which neglects the gap between natural language and symbolic structures.

**Graph Neural Networks** (GNNs) (Morris et al., 2019; Velickovic et al., 2017; Hamilton et al., 2017; Kipf and Welling, 2017) have been demonstrated to be effective in inducing and aggregating symbolic structures on other multi-hop question answering problems (Cao et al., 2019; Fang et al., 2020; Huang and Yang, 2021; Heo et al., 2022; Xu et al., 2021a; Deng et al., 2023b). In practice, the required number of graph layers grows with the multi-hop dependency between two distant nodes (Wang et al., 2021; Hong et al., 2022), which inevitably encounters the problem of over-smoothing (Li et al., 2018). Some researchers have conducted studies on relieving this problem (Wu et al., 2023; Min et al., 2020; Huang and Li, 2023; Yang et al., 2023; Liu et al., 2023; Koishekenov, 2023; Song et al., 2023). However, these methods are all breadth-aggregation-based, *i.e.*, they only posed adjustments on breadth aggregation like improving the aggregation filters, scattering the aggregation target using the probabilistic tool, etc., but never jumping out it. In this work, we investigate a depth-wise aggregation approach that captures long-range dependencies across any distance without the need to increase the model depth.

## 3 Method

### 3.1 Problem Definition

Following previous studies on spatial reasoning in text (Mirzaee et al., 2021; Shi et al., 2022; Mirzaee and Kordjamshidi, 2022), we define the problem as follows: Given a story description $S$ consisting of multiple sentences, the system aims to answer a question $Q$ based on the story $S$ by selecting a correct answer from the fixed set of given candidate answers regarding the spatial relations.

### 3.2 The DepWiNet

As presented in Figure 2, the overall framework named DepWiNet consists of three modules: the entity representation extraction module, the DepWiGNN reasoning module, and the prediction module. The entity representation extraction module provides comprehensive entity embedding that is used in the reasoning module. A graph with recognized entities as nodes is constructed after obtaining entity representations, which later will be fed into DepWiGNN. The DepWiGNN reasons over the constructed graph and updates the node embedding correspondingly. The final prediction module adds the entity embedding from DepWiGNN to the embeddings from the first extraction module and applies a single step of attention (Vaswani et al., 2017) to generate the final result.

**Entity Representation Extraction Module** We leverage PLMs to extract entity representations[1]. The model takes the concatenation of the story $S$ and the question $Q$ as the input and output embeddings of each token. The output embeddings are further projected using a single linear layer.

$$\hat{\mathbf{S}}, \hat{\mathbf{Q}} = \mathbf{PLM}(S, Q) \qquad (1)$$

$$\hat{\mathbf{S}}_\alpha = \hat{\mathbf{S}}\mathbf{W}_\alpha^T \quad \hat{\mathbf{Q}}_\alpha = \hat{\mathbf{Q}}\mathbf{W}_\alpha^T \qquad (2)$$

where $\mathbf{PLM}(\cdot)$ denotes the PLM-based encoder, *e.g.*, BERT (Devlin et al., 2019). $\hat{\mathbf{S}}_\alpha \in \mathbb{R}^{r_S \times d_h}$, $\hat{\mathbf{Q}}_\alpha \in \mathbb{R}^{r_Q \times d_h}$ denotes the list of projected tokens of size $d_h$ in the story and question, and $\mathbf{W}_\alpha \in \mathbb{R}^{d_h \times d_h}$ is the shared projection matrix. The entity representation is just the mean pooling of all token embeddings belonging to that entity.

**Graph Construction** The entities are first recognized from the input by employing rule-based entity recognition. Specifically, in StepGame (Shi et al., 2022), the entities are represented by a single capitalized letter, so we only need to locate all single capitalized letters. For SPARTUN

---

[1]Entity in this paper refers to the spatial roles defined in (Kordjamshidi et al., 2010).

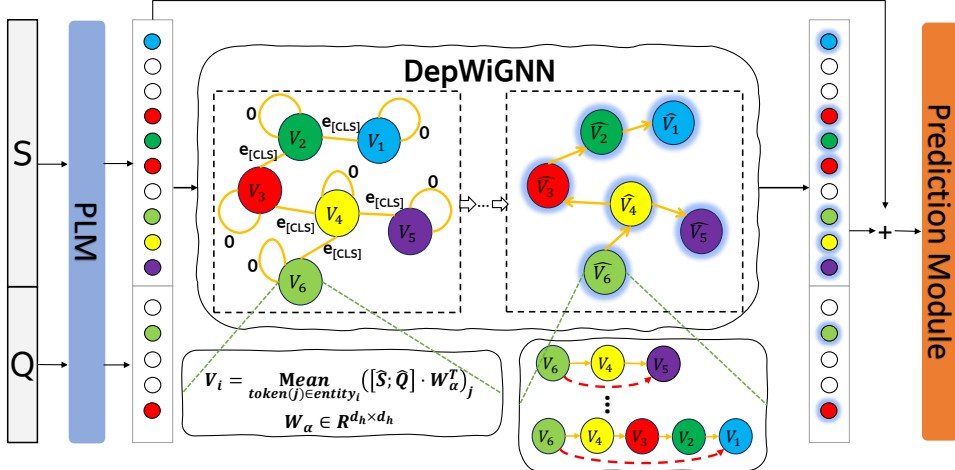

Figure 2: The DepWiNet framework. The entity representations are first extracted from the entity representation extraction module (left), and then a homogeneous graph is constructed based on the entity embeddings and fed into the DepWiNet reasoning module. The DepWiNet depth-wisely aggregates information for all indirectly connected node pairs, and stores it in node memories. The updated node embeddings are then passed to the prediction module.

and ReSQ, we use nltk RegexpParser[2] with self-designed grammars[3] to recognize entities. Entities and their embeddings are treated as nodes of the graph, and an edge between two entities exists if and only if the two entities co-occur in the same sentence. We also add an edge feature for each edge.

$$E_{ij} = \begin{cases} \mathbf{0} & \text{if i == j,} \\ \mathbf{e}_{[CLS]} & \text{ortherwise.} \end{cases} \quad (3)$$

If two entities are the same (self-loop), the edge feature is just a zero tensor with a size $d_h$; otherwise, it is the sequence's last layer hidden state of the $[CLS]$ token. The motivation behind treating [CLS] token as an edge feature is that it can help the model better understand the atomic relation (k=1) between two nodes (entities) so as to facilitate later depth aggregation. A straightforward justification can be found in Table 1, where all three PLMs achieve very high accuracy at k=1 cases by using the [CLS] token, which demonstrates that the [CLS] token favors the understanding of the atomic relation.

**DepWiGNN Module** The DepWiGNN module (middle part in Figure 2) is responsible for multi-hop reasoning and can collect arbitrarily distant dependency without layer stacking.

$$\hat{V} = \mathbf{DepWiGNN}(\mathcal{G}; V, E) \quad (4)$$

[2]https://www.nltk.org/api/nltk.chunk.regexp.html
[3]r"""NP:<ADJ|NOUN>+<NOUN|NUM>+""" and r"""NP:<ADJ|NOUN>+<VERB>+<NOUN|NUM>"""

where $V \in \mathbb{R}^{|V| \times d_h}$ and $E \in \mathbb{R}^{|E| \times d_h}$ are the nodes and edges of the graph. It comprises three components: node memory initialization, long dependency collection, and spatial relation retrieval. Details of these three components will be discussed in Section 3.3.

**Prediction Module** The node embedding updated in DepWiGNN is correspondingly added to entity embedding extracted from PLM.

$$\hat{Z}_i = \begin{cases} Z_i + \hat{V}_{idx(i)} & \text{if } i\text{-th token} \in \hat{V} \\ Z_i & \text{ortherwise.} \end{cases} \quad (5)$$

where $Z_i = [\hat{S}; \hat{Q}]$ is the ith token embedding from PLM and $\hat{V}$ denotes the updated entity representation set. $idx(i)$ is the index of $i$-th token in the graph nodes.

Then, the sequence of token embeddings in the question and story are extracted separately to perform attention mechanism (Vaswani et al., 2017) with the query to be the sequence of question tokens embeddings $\hat{Z}_{\hat{Q}}$, key and value to be the sequence of story token embeddings $\hat{Z}_{\hat{S}}$.

$$R = \sum_{i=0}^{r_Q} (softmax(\frac{\hat{Z}_{\hat{Q}} \hat{Z}_{\hat{S}}^T}{\sqrt{d_h}})\hat{Z}_{\hat{S}})_i \quad (6)$$

$$C = FFN(\mathbf{layernm}(R)) \quad (7)$$

where **layernm** means layer normalization and $C$ is the final logits. The result embeddings are summed over the first dimension, layernormed, and

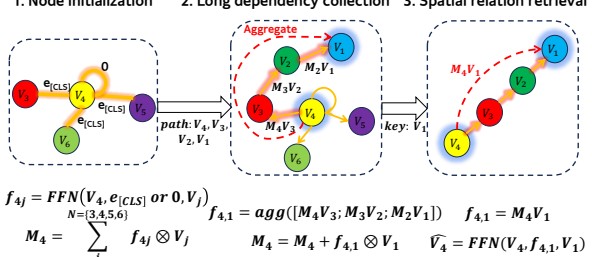

$$f_{4j} = FFN(V_4, e_{[CLS]} \text{ or } 0, V_j)$$
$$\underset{N=\{3,4,5,6\}}{M_4 = \sum_j f_{4j} \otimes V_j}$$
$$f_{4,1} = agg([M_4V_3; M_3V_2; M_2V_1]) \quad f_{4,1} = M_4V_1$$
$$M_4 = M_4 + f_{4,1} \otimes V_1 \quad \hat{V}_4 = FFN(V_4, f_{4,1}, V_1)$$

Figure 3: The illustration of DepWiGNN.

fed into a 3-layer feedforward neural network to acquire the final result. The overall framework is trained in an end-to-end manner to minimize the cross-entropy loss between the predicted candidate answer probabilities and the ground-truth answer.

### 3.3 Depth-wise Graph Neural Network

As illustrated in Figure 3, we introduce the proposed graph neural network, called DepWiGNN, *i.e.*, the operation $\hat{V} = \mathbf{DepWiGNN}(\mathcal{G}; V, E)$. Unlike existing GNNs (e.g. (Morris et al., 2019; Velickovic et al., 2017; Hamilton et al., 2017)), which counts the one-dimension node embedding itself as its memory to achieve the function of information reservation, updating, and retrieval, DepWiGNN employs a novel two-dimension node memory implementation approach that takes the advantage of TPR mechanism allowing the updating and retrieval operations of the memory to be conveniently realized by simple arithmetic operations like plus, minus and outer product. This essentially determines that the information propagation between any pair of nodes with any distance in the graph can be accomplished without the demand to iteratively apply neural layers.

**Node Memory Initialization** At this stage, the node memories of all nodes are initialized with the relations to their immediate neighbors. In the multi-hop spatial reasoning cases, the relations will be the atomic spatial orientation relations (which only needs one hop) of the destination node relative to the source node. For example, "X is to the left of K and is on the same horizontal plane." We follow the TPR mechanism (Smolensky, 1990), which uses outer product operation to bind roles and fillers[4]. In this work, the calculated spatial vectors are considered to be the fillers. They will be bound with corresponding node embeddings and stored in the two-dimensional node memory. Explicitly, we first

---

[4]The preliminary of TPR is presented in Appendix A.

acquire spatial orientation filler $f_{ij} \in \mathbb{R}^{d_h}$ by using a feedforward network, the input to FFN is concatenation in the form $[V_i, E_{ij}, V_j]$. $V_i$ represents $i$-th node embedding.

$$f_{ij} = FFN([V_i, E_{ij}, V_j]) \tag{8}$$
$$M_i = \sum_{V_j \in N(V_i)} f_{ij} \otimes V_j \tag{9}$$

The filler is bound together with the corresponding destination node $V_j$ using the outer product. The initial node memory $M_i$ for node $V_i$ is just the summation of all outer product results of the fillers and corresponding neighbors (left part in Figure 3).

**Long Dependency Collection** We discuss how the model collects long dependencies in this section. Since the atomic relations are bound by corresponding destinations and have already been contained in the node memories, we can easily unbind all the atomic relation fillers in a path using the corresponding atomic destination node embedding (middle part in Figure 3). For each pair of indirectly connected nodes, we first find the shortest path between them using breadth-first search (BFS). Then, all the existing atomic relation fillers along the path are unbound using the embedding of each node in the path (Eq.10).

$$\hat{f}_{p_i(p_{i+1})} = M_{p_i} V_{p_{i+1}}^T \tag{10}$$
$$\hat{\mathbf{F}} = \mathbf{Aggregator}(\mathbf{F}) \tag{11}$$
$$f_{p_0 p_n} = \mathbf{layernm}(FFN(\hat{\mathbf{F}}) + \hat{\mathbf{F}}) \tag{12}$$
$$M_{p_0} = M_{p_0} + f_{p_0 p_n} \otimes V_{p_n}^T \tag{13}$$

where $p_i$ denotes the $i$-th element in the path and $\mathbf{F} = [\hat{f}_{p_0 p_1}; ....; \hat{f}_{p_{n-1} p_n}]$ is the retrived filler set along the path. The collected relation fillers are aggregated using a selected depth aggregator like LSTM (Hochreiter and Schmidhuber, 1997), etc, and passed to a feedforward neural network to reason the relation filler between the source and destination node in the path (Eq.12). The result spatial filler is then bound with the target node embedding and added to the source node memory (Eq.13). In this way, each node memory will finally contain the spatial orientation information to every other connected node in the graph.

**Spatial Relation Retrieval** After the collection process is completed, every node memory contains spatial relation information to all other connected nodes in the graph. Therefore, we can conveniently retrieve the spatial information from a source node

to a target node by unbinding the spatial filler from the source node memory (right part in Figure 3) using a self-determined key. The key can be the target node embedding itself if the target node can be easily recognized from the question, or some computationally extracted representation from the sequence of question token embeddings if the target node is hard to discern. We use the key to unbind the spatial relation from all nodes' memory and pass the concatenation of it with source and target node embeddings to a multilayer perceptron to get the updated node embeddings.

$$\hat{f}_{i[key]} = M_i V_{[key]}^T \tag{14}$$

$$\hat{V}_i = FFN([V_i, \textbf{layernm}(f_{i[key]}), V_{[key]}]) \tag{15}$$

The updated node embeddings are then passed to the prediction module to get the final result.

## 4 Experiments

### 4.1 Experimental Setups

**Datasets & Evaluation Metrics** We investigated our model on StepGame (Shi et al., 2022) and ReSQ (Mirzaee and Kordjamshidi, 2022) datasets, which were recently published for multi-hop spatial reasoning. StepGame is a synthetic textual QA dataset that has a number of relations ($k$) ranging from 1 to 10. In particular, we follow the experimental procedure in the original paper (Shi et al., 2022), which utilized the TrainVersion of the dataset containing 10,000/1,000 training/validation clean samples for each $k$ value from 1 to 5 and 10,000 noisy test examples for $k$ value from 1 to 10. The test set contains three kinds of distraction noise: *disconnected*, *irrelevant*, and *supporting*. ReSQ is a crowdsourcing benchmark that includes only Yes/No questions with 1008, 333, and 610 examples for training, validating, and testing respectively. Since it is human-generated, it can reflect the natural complexity of real-world spatial description. Following the setup in (Shi et al., 2022; Mirzaee and Kordjamshidi, 2022), we report the accuracy on corresponding test sets for all experiments.

**Baselines** For StepGame, we select all traditional reasoning models used in the original paper (Shi et al., 2022) and three PLMs, *i.e.*, BERT (Devlin et al., 2019), RoBERTa (Liu et al., 2019), and ALBERT (Lan et al., 2020) as our baselines. For ReSQ, we also follow the experiment setting described in (Mirzaee and Kordjamshidi, 2022),

which used BERT with or without further synthetic supervision as baselines.

**Implementation Details** For all experiments, we use the base version of corresponding PLMs, which has 768 embedding dimensions. The model was trained in an end-to-end manner using Adam optimizer (Kingma and Ba, 2015). The training was stopped if, up to 3 epochs, there is no improvement greater than 1e-3 on the cross-entropy loss for the validation set. We also applied a Pytorch training scheduler that reduces the learning rate with a factor of 0.1 if the improvement of cross-entropy loss on the validation set is lower than 1e-3 for 2 epochs. In terms of the determination of the key in the Spatial Relation Retrieval part, we used the target node embedding for StepGame since it can be easily recognized, and we employed a single linear layer to extract the key representation from the sum-aggregated question token embeddings for ReSQ. In the StepGame experiment, we fine-tune the model on the training set and test it on the test set. For ReSQ, we follow the procedure in (Mirzaee and Kordjamshidi, 2022) to test the model on ReSQ with or without further supervision from SPARTUN (Mirzaee and Kordjamshidi, 2022). Unless specified, all the experiments use LSTM (Hochreiter and Schmidhuber, 1997) as depth aggregator by default. The detailed hyperparameter settings are given in the Appendix B.

### 4.2 Overall Performance

Table 1 and 2 report the experiment results on StepGame and ReSQ respectively. As shown in Table 1, PLMs overwhelmingly outperform all the traditional reasoning models and the proposed DepWiNet overtakes the PLMs by a large margin, especially for the cases with greater $k$ where the multi-hop reasoning capability plays a more important role. This aligns with the characteristics of our model architecture that the aggregation focuses on the depth dimension, which effectively avoids the problem of over-smoothing and the mixture of redundant information from the breadth aggregation. Despite the model being only trained on clean distraction-noise-absent samples with $k$ from 1 to 5, it achieves impressive performance on the distraction-noise-present test data with $k$ value from 6 to 10, demonstrating the more advanced generalization capability of our model. Moreover, our method also entails an improvement for the examples with lower $k$ values for BERT and AL-

| Method | k=1 | k=2 | k=3 | k=4 | k=5 | k=6 | k=7 | k=8 | k=9 | k=10 | Mean(k=1-5) | Mean(k=6-10) |
|---|---|---|---|---|---|---|---|---|---|---|---|---|
| RN (Santoro et al., 2017) | 22.64 | 17.08 | 15.08 | 12.84 | 11.52 | 11.12 | 11.53 | 11.21 | 11.13 | 11.34 | 15.83 | 11.27 |
| RRN (Palm et al., 2018) | 24.05 | 19.98 | 16.03 | 13.22 | 12.31 | 11.62 | 11.40 | 11.83 | 11.22 | 11.69 | 17.12 | 11.56 |
| UT (Dehghani et al., 2019) | 45.11 | 28.36 | 17.41 | 14.07 | 13.45 | 12.73 | 12.11 | 11.40 | 11.41 | 11.74 | 23.68 | 11.88 |
| STM (Le et al., 2020) | 53.42 | 35.96 | 23.03 | 18.45 | 15.14 | 13.80 | 12.63 | 11.54 | 11.30 | 11.77 | 29.20 | 12.21 |
| TPR-RNN (Schlag and Schmidhuber, 2018b) | 70.29 | 46.03 | 36.14 | 26.82 | 24.77 | 22.25 | 19.88 | 15.45 | 13.01 | 12.65 | 40.81 | 16.65 |
| TP-MANN (Shi et al., 2022) | 85.77 | 60.31 | 50.18 | 37.45 | 31.25 | 28.53 | 26.45 | 23.67 | 22.52 | 21.46 | 52.99 | 24.53 |
| BERT (Mirzaee and Kordjamshidi, 2022) | 98.53 | 93.40 | 91.19 | 66.98 | 54.57 | 48.59 | 42.81 | 37.98 | 34.16 | 33.62 | 80.93 | 39.43 |
| RoBERTa* (Mirzaee and Kordjamshidi, 2022) | 98.68 | 97.05 | 95.66 | 79.59 | 74.89 | 70.67 | 66.01 | 61.42 | 57.20 | 54.53 | 89.17 | 61.93 |
| ALBERT* (Mirzaee and Kordjamshidi, 2022) | 98.56 | 97.56 | 96.08 | 90.01 | 83.33 | 77.24 | 71.57 | 64.47 | 60.02 | 56.18 | 93.11 | 65.90 |
| **DepWiNet** (BERT) | 98.44 | 96.57 | 94.75 | 81.41 | 70.68 | 62.80 | 57.46 | 49.14 | 45.56 | 44.01 | $88.37_{+9.2\%}$ | $51.79_{+31.3\%}$ |
| **DepWiNet** (RoBERTa) | 98.70 | 96.69 | 95.54 | 79.50 | 74.94 | 70.37 | 66.22 | 61.89 | 58.01 | 56.89 | $89.07_{-0.1\%}$ | $62.68_{+1.2\%}$ |
| **DepWiNet** (ALBERT) | 98.59 | 97.13 | 95.87 | 91.96 | 87.50 | 82.62 | 77.80 | 71.40 | 67.55 | 64.98 | $94.21_{+1.2\%}$ | $72.87_{+10.6\%}$ |

Table 1: Experimental results on StepGame. * denotes the implementation of the same model as BERT (Mirzaee and Kordjamshidi, 2022) with RoBERTa and ALBERT. The other results are reported in (Shi et al., 2022).

| Method | *SynSup* | ReSQ |
|---|---|---|
| Majority Baseline | - | 50.21 |
| BERT | - | 57.37 |
| BERT | SPARTQA-AUTO | 55.08 |
| BERT | StepGame | 60.14 |
| BERT | SPARTUN-S | 58.03 |
| BERT | SPARTUN | 63.60 |
| **DepWiNet** (BERT) | - | **63.30** |
| **DepWiNet** (BERT) | SPARTUN | **65.54** |

Table 2: Experimental results on ReSQ, under the transfer learning setting (Mirzaee and Kordjamshidi, 2022).

BERT and an ignorable decrease for RoBERTa, which attests to the innocuity of our model to the few-hop reasoning tasks.

Experimental results in Table 2 show that DepWiNet reaches a new SOTA result on ReSQ for both cases where extra supervision from SPARTUN is present and absent. Excitingly, the performance of our model without extra supervision even overwhelms the performance of the BERT with extra supervision from SPARTQA-AUTO (Mirzaee et al., 2021), StepGame, and SPARTUN-S and approaches closely to the SPARTUN supervision case. All these phenomenons signify that our model has the potential to better tackle the natural intricacy of real-world spatial expressions.

## 4.3 Ablation study

We conduct ablation studies on the impact of the three components of DepWiGNN and different depth aggregators, as presented in Table 3.

**Impact of DepWiNet components** The model performance experiences a drastic decrease, particularly for the mean score of $k$ between 6 and 10, after the Long Dependency Collection (LDC) has been removed, verifying that this component serves a crucial role in the model. Note that the mean score for $k$(6-10) even drops below the ALBERT baseline (Table 1). This is reasonable as the LDC is directly responsible for collecting long dependencies. We further defunctionalized the Node Memory Initialization (NMI) and then Spatial Relation Retrieval (SRR) by setting the initial fillers (Eq.8) and key representation (Eq.14) to a random vector separately. Compared with the case where only LDC was removed, both of them lead to a further decrease in small and large $k$ values.

**Impact of Different Aggregators** The results show that the mean and max pooling depth aggregators fail to understand the spatial rule as achieved by LSTM. This may be caused by the relatively less expressiveness and generalization caliber of mean and max pooling operation.

## 4.4 Comparisons with Different GNNs

To certify our model's capacity of collecting long dependencies as well as its immunity to the over-smoothing problem, we contrast it with four graph neural networks, namely, GCN (Kipf and Welling, 2017), GraphConv (Morris et al., 2019), GAT (Velickovic et al., 2017) and GCNII (Chen et al., 2020c). We consider the cases with the number of layers varying from 1 to 5 and select the best performance to compare. The plot of the accuracy metric is reported in Figure 4 and the corresponding best

| Method | k=1 | k=2 | k=3 | k=4 | k=5 | k=6 | k=7 | k=8 | k=9 | k=10 | Mean(k=1-5) | Mean(k=6-10) |
|---|---|---|---|---|---|---|---|---|---|---|---|---|
| DepWiNet (ALBERT) | **98,59** | **97.13** | **95.87** | **91.96** | **87.50** | **82.62** | **77.80** | **71.40** | **67.55** | **64.98** | **94.21** | **72.87** |
| - w/o LDC | 98.58 | 97.72 | 96.42 | 82.46 | 78.53 | 74.52 | 70.45 | 62.84 | 59.06 | 56.57 | $90.74_{-3.7\%}$ | $64.69_{-11.2\%}$ |
| - w/o NMI & LDC | 98.48 | 96.88 | 94.83 | 79.14 | 75.46 | 70.51 | 67.19 | 61.88 | 57.65 | 55.02 | $88.96_{-5.6\%}$ | $62.45_{-14.3\%}$ |
| - w/o SRR & LDC | 98.64 | 97.52 | 95.91 | 80.72 | 76.85 | 72.56 | 67.93 | 61.29 | 57.97 | 54.17 | $89.93_{-4.5\%}$ | $62.78_{-13.8\%}$ |
| - w/ DepWiGNN$_{mean}$ | 98.52 | 96.46 | 93.69 | 81.10 | 77.90 | 73.47 | 70.78 | 65.54 | 62.42 | 59.82 | $89.53_{-5.0\%}$ | $66.41_{-8.9\%}$ |
| - w/ DepWiGNN$_{maxpooling}$ | 98.65 | 95.83 | 93.90 | 80.81 | 77.20 | 72.07 | 68.76 | 62.79 | 59.63 | 57.14 | $89.28_{-5.2\%}$ | $64.08_{-12.1\%}$ |

Table 3: Ablation study. NMI, LDC, and SRR denotes the three stages in DepWiGNN, *i.e.*, Node Memory Initialization, Long Dependency Collection, and Spatial Relation Retrieval, respectively.

| Method | k=1 | k=2 | k=3 | k=4 | k=5 | k=6 | k=7 | k=8 | k=9 | k=10 | Mean(k=1-5) | Mean(k=6-10) |
|---|---|---|---|---|---|---|---|---|---|---|---|---|
| ALBERT | 98.56 | 97.56 | 96.08 | 90.01 | 83.33 | 77.24 | 71.57 | 64.47 | 60.02 | 56.18 | 93.11 | 65.90 |
| w/ GCN$_5$ (Kipf and Welling, 2017) | 98.62 | 97.72 | **96.49** | 82.01 | 76.70 | 71.60 | 67.34 | 60.49 | 56.11 | 52.65 | $90.31_{-3.01\%}$ | $61.64_{-6.46\%}$ |
| w/ GraphConv$_2$ (Morris et al., 2019) | 98.66 | **97.73** | 96.48 | 82.37 | 78.57 | 74.47 | 69.91 | 63.04 | 60.01 | 56.58 | $90.76_{-2.52\%}$ | $64.80_{-1.67\%}$ |
| w/ GAT$_1$ (Velickovic et al., 2017) | **98.67** | 97.34 | 95.59 | 81.47 | 78.39 | 73.79 | 70.17 | 63.52 | 60.64 | 57.33 | $90.29_{-3.03\%}$ | $65.09_{-1.23\%}$ |
| w/ GCNII$_4$ (Chen et al., 2020c) | **98.56** | 97.61 | 96.36 | 83.31 | 79.22 | 74.88 | 70.64 | 63.64 | 60.59 | 56.99 | $91.01_{-2.25\%}$ | $65.35_{+0.01\%}$ |
| w/ DepWiGNN | 98.59 | 97.13 | 95.87 | **91.96** | **87.50** | **82.62** | **77.80** | **71.40** | **67.55** | **64.98** | **94.21**$_{+1.18\%}$ | **72.87**$_{+10.58\%}$ |

Table 4: Comparisons with different GNNs. The subscripts represent the number of GNN layers. We select the best mean performance among layer number 1 to 5.

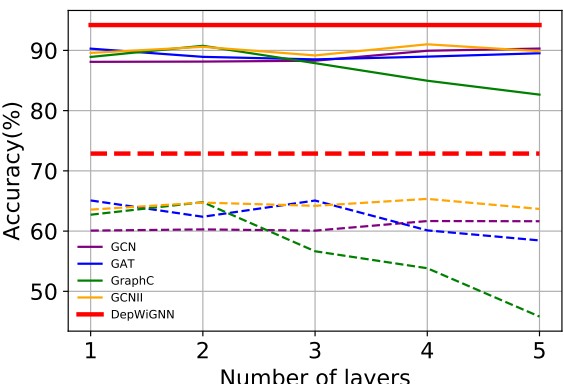

Figure 4: Impact of the layer number in different GNNs on StepGame. The solid and dashed lines denote the mean score of ($k$=1-5) and ($k$=6-10) respectively.

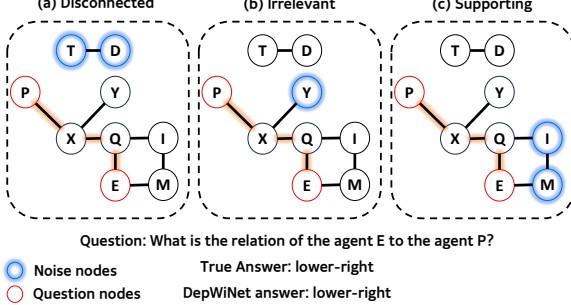

Figure 5: Cases with distracting noisy from StepGame.

cases are in Table 4. It is worth noting that almost all the baseline GNNs cause a reduction in the original PLM performance (Table 4). The reason for this may partially come from the breadth aggregation, which aggregates the neighbors round and round and leads to indistinguishability among entity embeddings such that the PLM reasoning process has been disturbed. The majority of baseline GNNs suffer from an apparent performance drop when increasing the number of layers (Figure 4), while our model consistently performs better and is not affected by the number of layers at all since it does not use breadth aggregation. Therefore, our model has immunity to over-smoothing problems.

In both small and large $k$ cases, our model outperforms the best performance of all four GNNs (including GCNII, which is specifically designed for over-smoothing issues) by a large margin (Table 4), which serves as evidence of the superiority of our model in long dependency collection.

### 4.5 Case study

In this section, we present case studies to intuitively show how DepWiGNN mitigates the three kinds of distracting noise introduced in StepGame, namely, *disconnected*, *irrelevant*, and *supporting*.

• The *disconnected* noise is the set of entities and relations that forms a new independent chain in the graph (Figure 5(a)). The node memories constructed in DepWiGNN contain spatial information about the nodes if and only if that node stays in the same connected component; otherwise, it

has no information about the node as there is no path between them. Hence, in this case, for the questioned source node **P**, its memory has no information for the disconnected noise **T** and **D**.

- The *irrelevant* noise branches the correct reasoning chain out with new entities and relations but results in no alternative reasoning path (Figure 5(b)). Hence the irrelevant noise entities will not be included in the reasoning path between the source and destination, which means that it will not affect the destination spatial filler stored in the source node memory. In this case, when the key representation (embedding of entity **E**) is used to unbind the spatial filler from node memory of the source node **P**, it obtains the filler $f_{PE} = FFN(Aggregator([f_{PX}; f_{XQ}; f_{QE}]))$, which is intact to the irrelevant entity **Y** and relation $f_{xy}$ or $f_{yx}$.

- The *supporting* noise adds new entities and relations to the original reasoning chain that provides an alternative reasoning path (Figure 5(c)). DepWiGNN is naturally exempted from this noise for two reasons: first, it finds the shortest path between two entities, therefore, will not include **I** and **M** in the path; Second, even if the longer path is considered, the depth aggregator should reach the same result as the shorted one since the source and destination are the same.

## 5 Conclusion

In this work, we introduce DepWiGNN, a novel graph-based method that facilitates depth-wise propagation in graphs, enabling effective capture of long dependencies while mitigating the challenges of over-smoothing and excessive layer stacking. Our approach incorporates a node memory scheme leveraging the TPR mechanism, enabling simple arithmetic operations for memory updating and retrieval, instead of relying on additional neural layers. Experiments on two recently released textual multi-hop spatial reasoning datasets demonstrate the superiority of DepWiGNN in collecting long dependencies over the other three typical GNNs and its immunity to the over-smoothing problem.

## Limitation

Unlike the classical GNNs, which use one-dimensional embedding as the node memory, our method applies a two-dimensional matrix-shaped node memory. This poses a direct increase in memory requirement. The system has to assign extra space to store a matrix with shape $\mathbb{R}^{d_h \times d_h}$ for each node in the graph, which makes the method less scalable. However, it is a worthwhile trade-off because the two-dimensional node memory can potentially store $d_h - 1$ number of spatial fillers bound with node embeddings while keeping the size fixed, and it does not suffer from information overloading as faced by the one-dimension memory since the spatial fillers can be straightforwardly retrieved from the memory.

Another issue is the time complexity of finding the shortest path between every pair of nodes. For all experiments in this paper, since the edges do not have weights, the complexity is O((n+e)*n), where n and e are the numbers of nodes and edges, respectively. However, things will get worse if the edges in the graph are weighted. We believe this is a potential future research direction for the improvement of the algorithm.

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

# Appendix

## A   Preliminary of Tensor Product Representation

Consider the implicit relation hidden in the sentence "X is over K." Our method decomposes this sentence into two groups of role $r \in \mathcal{R}$ and filler $f \in \mathcal{F}$ symbol components, i.e., $\mathcal{R} = \{X, K\}$ and $\mathcal{F} = \{above, below\}$. The role and filler symbols are projected into role $V_{\mathcal{R}}$ and filler $V_{\mathcal{F}}$ vector space, respectively. The TPR of these symbol structures is defined as the tensor $\mathbf{T}$ within the vector space $V_{\mathcal{R}} \otimes V_{\mathcal{F}}$, where $\otimes$ denotes the tensor product (equivalent to outer product when role and filler are one-dimension vectors). In this example, we can drive the equation of writing our example to $\mathbf{T}$:

$$\begin{aligned}
\mathbf{T} &= f_{above} \otimes r_X + f_{below} \otimes r_K \\
&= f_{above} r_X^T + f_{below} r_K^T
\end{aligned} \quad (16)$$

The tensor product $\otimes$ acts as a binding operator to bind the corresponding role and filler.

The tensor inner product serves as the unbinding operator in TPR, which is employed to reconstruct the previously reserved fillers from $\mathbf{T}$. Given the

| Experiment | Dataset | epochs | batch size | PLM learning rate | non-PLM learning rate |
|---|---|---|---|---|---|
| BERT | StepGame | 22 | 32 | 2e-05 | - |
| ALBERT | StepGame | 14 | 32 | 6e-06 | - |
| RoBERTa | StepGame | 16 | 32 | 6e-06 | - |
| DepWiNet(BERT) | StepGame | 17 | 32 | 2e-05 | 1e-04 |
| DepWiNet(ALBERT) | StepGame | 22 | 32 | 6e-06 | 5e-05 |
| DepWiNet(RoBERTa) | StepGame | 19 | 32 | 6e-06 | 1e-04 |
| DepWiNet(BERT) | SPARTUN | 9 | 16 | 8e-06 | 1e-04 |
| DepWiNet(BERT) | ReSQ(with SPARTUN supervision) | 4 | 16 | 5e-05 | 4e-05 |
| DepWiNet(BERT) | ReSQ(without SPARTUN supervision) | 4 | 16 | 5e-05 | 4e-05 |
| DepWiNet(w/o LDC) | StepGame | 15 | 32 | 6e-06 | 5e-05 |
| DepWiNet(w/o NMI & LDC) | StepGame | 14 | 32 | 6e-06 | 5e-05 |
| DepWiNet(w/o SRR & LDC) | StepGame | 14 | 32 | 6e-06 | 5e-05 |
| DepWiNet(w/ DepWiGNN$_{mean}$) | StepGame | 14 | 32 | 6e-06 | 5e-05 |
| DepWiNet(w/ DepWiGNN$_{maxpooling}$) | StepGame | 14 | 32 | 6e-06 | 5e-05 |
| DepWiNet(w/ GCN$_{1-5}$) | StepGame | (16,12,10,19,14) | 32 | 6e-06 | 5e-05 |
| DepWiNet(w/ GraphConv$_{1-5}$) | StepGame | (14,14,16,20,15) | 32 | 6e-06 | 5e-05 |
| DepWiNet(w/ GAT$_{1-5}$) | StepGame | (16,13,16,16,17) | 32 | 6e-06 | 5e-05 |

Table 5: Hyperparameters and setup information for each experiment.

unbinding vector $u_K$, we can retrieve the stored filler $f_{below}$ from the $\mathbf{T}$. In the most ideal cases, if the role vectors are orthogonal to each other, $u_K$ equals $r_K$. When $\mathbf{T} \in \mathbb{R}^2$, it can be expressed as the matrix multiplication.:

$$\begin{aligned} \mathbf{T} \cdot u_K &= (f_{above}r_X^T + f_{below}r_K^T) \cdot r_K \\ &= \alpha f_{below} \propto f_{below} \end{aligned} \quad (17)$$

## B  Hyperparameter Settings

Detailed hyperparameter settings for each experiment are provided in Table 5. Epoch numbers for DepWiNet with the three classical GNNs are grouped (in order) for simplicity.