# OpenReview forum: "DepWiGNN: A Depth-wise Graph Neural Network for Multi-hop Spatial Reasoning in Text"
_EMNLP/2023/Conference — EMNLP 2023 Findings_

### Official Review · Reviewer_b7C1 · 2023-08-02

**Soundness:** 4

**Excitement:**

4: Strong: This paper deepens the understanding of some phenomenon or lowers the barriers to an existing research direction.

**Paper Topic And Main Contributions:**

The paper proposes a novel Depth-Wise Graph Neural Network (DepWiGNN) for spatial reasoning in text. It addresses the challenges faced by classical GNNs in handling multi-hop spatial reasoning, specifically the over-smoothing issue, by designing a unique node memory scheme and aggregating information over the depth dimension of the graph. The proposed Transformed Product Rule (TPR) mechanism enables convenient memory updating and retrieval operations through simple arithmetic operations, rather than relying on additional neural layers. The authors claim thate xperimental results on challenging multi-hop spatial reasoning datasets demonstrate that DepWiGNN outperforms existing spatial reasoning methods.




**Reasons To Accept:**

- The authors introduce a novel GNN modeling approach that incorporates a node memory scheme that leverages the TPR mechanism, enabling simple arithmetic operations for memory updating and retrieval, instead of relying on additional neural layers.

- The authors perform several experiments and benchmark with both popular LLMs and GNNs to validate the performance of their approach.


**Reasons To Reject:**

- Some experiments do not look very comprehensive to me, for instance, why don't the authors experiment with breath based GNNs in the StepGame experiment?

- The authors constantly highlight the fact that DepWiGNN doesn't encounter the oversmoothing issue like regular GNNs but do not show empirical evidence for the same.

**Reproducibility:**

4: Could mostly reproduce the results, but there may be some variation because of sample variance or minor variations in their interpretation of the protocol or method.

**Reviewer Confidence:**

4: Quite sure. I tried to check the important points carefully. It's unlikely, though conceivable, that I missed something that should affect my ratings.

---

> ### Author Rebuttal · Authors · 2023-08-28
>
> Thank you for your comments.
>
> We, indeed, conduct experiments to verify the over-smoothing issue. Following the standard experimental settings in previous studies [1,2], Section 4.4 includes experiments for breadth based GNNs (GCN, GAT, GraphC) on the StepGame dataset, and it serves as the empirical evidence that our model alleviates the problem of over-smoothing. Specifically, we compared our model against the three breadth based GNNs on various layer numbers from 1-5. The result shows that the performance of our model surpasses all other baselines by a large margin for all layer number cases. This proves two things: First, the typical GNNs do suffer from the over-smoothing problem because their performance drops evidently while the number of layers increases (especially for the large k values); Second, our model has better immunity to the problem because, theoretically, it does not need to stack multiple layers to collect long dependencies and, empirically, it consistently performs better than all the other models.
>
> We also followed the suggestion from another reviewer and did one extra experiment for the model GCNII [3], which is a popular model designed to overcome over-smoothing problems. The results are shown below:
>
> | Mean(k=1-5)/(k=6-10)  | L=1  | L=2  | L=3  | L=4  | L=5  |
> |---|---|---|---|---|---|
> | GCNII | 89.58/63.57  |  90.59/64.73 | 89.17/64.20  | 91.01/65.35  |  89.87/63.68 |
> | DepWiGNN  | 94.21/72.87  | 94.21/72.87  | 94.21/72.87  | 94.21/72.87  | 94.21/72.87  |
>
> L means the number of layers.
>
> While the GCNII performance does not decline too much with increasing layer number, it underperforms our model by a large margin. This also proves our model’s immunity to the over-smoothing problem. We will add this result and further highlight the analysis of the over-smoothing issue to address your concern better.
>
> - [1] Li, Qimai, Zhichao Han, and Xiao-Ming Wu. "Deeper insights into graph convolutional networks for semi-supervised learning." Proceedings of the AAAI conference on artificial intelligence. Vol. 32. No. 1. 2018.
> - [2] Wu, Gongce, et al. "QPGCN: Graph convolutional network with a quadratic polynomial filter for overcoming over-smoothing." Applied Intelligence 53.6 (2023): 7216-7231.
> - [3] Chen, Ming, et al. "Simple and deep graph convolutional networks." International conference on machine learning. PMLR, 2020.

---

### Official Review · Reviewer_YL2U · 2023-08-04

**Soundness:** 3

**Excitement:**

3: Ambivalent: It has merits (e.g., it reports state-of-the-art results, the idea is nice), but there are key weaknesses (e.g., it describes incremental work), and it can significantly benefit from another round of revision. However, I won't object to accepting it if my co-reviewers champion it.

**Paper Topic And Main Contributions:**

In this paper, the authors study the problem of multi-hop spatial reasoning for text. They introduce a limitation that plague GNN models when applied to
this task. To address these issue they propose a framework that can better collect long-term dependencies between two nodes in the graph. They detail their results
on multiple benchmarks. They further show that their model excels for questions of longer length (i.e., relations). Overall, while I find the results quite promising, I believe the motivation and writing need to be further improved at this stage.

Rebuttal Update: Raised my soundness score from a 2 &rarr; 3 after a discussion with the authors.

**Questions For The Authors:**

1. What is the definition of an "entity"? I don't see it formally defined in the paper.
2. In Section 3.1 the authors state "The entities are first recognized from the input by employing rule-based entity recognition.". Can you give more details.
I'm confused as to exactly how the entities are identified and their tokens combined.
3. Why is the [CLS] token used as the edge feature? What's the motivation/justification?
4. What's the complexity of DepWiGNN? While it avoid propagating for many layers, the BFS search and the long dependency collection seem to have a large impact on the efficiency. A complexity or runtime analysis would be appropriate.

**Reasons To Accept:**

1. The motivation is clear. They clearly state why traditiona GNNs may struggle in this task and why they design an alternative.
2. The main results on the baselines are promising. They show that their framework can achieve both strong overall performance, and good performance on
questions with longer dependencies.
3. The additional studies (ablation and case study) are useful and do a good job of supporting the main results.

**Reasons To Reject:**

1. The methodology is confusing at times (see questions for some clarification). The issue for me is that the details of the framework are just presented to the reader, instead of being better introduced and motivated. I'd recommend re-organizing Section 3, empahsizing (a) the challanges of designing a GNN that addresses your motivation, (b) what general ideas/components are required to meet these challenges, and (c) mention how each component you introduce can address a challenge. Putting this in the beginning of section 3 will give the reader a better context behind the design of your framework. As of now, a lot of the design seems arbitrary (e.g. see question 3 later).

2. While the motivation of the paper is clear (oversmoothing is detrimental to capturing long-term dependecies), more evidence should be given to support it. Specifically, the authors mention after introducing some works that study oversmoothing (on line 202) that "However, these methods are all breadth-aggregation-based ... but never jumping out of it". I think a clearer motivtion needs to be given why it's needed to "jump out" of the standard aggregation procedure. Is basic aggregation inheritely incapable of capturing the necessary information (theoretically or empirically)?


3. In Table and Figure 4, DepWiGNN is compared against other traditional GNN frameworks. It's clear that such frameworks struggle as k increases. However, such models are well known to suffer from over-smoothing. It would be better to include methods meant to address over-smoothing. I recommend including GCNII [1], a popular method for doing so. It's possible that it may fair better as k increases.


[1] Chen, Ming, et al. "Simple and deep graph convolutional networks." International conference on machine learning. PMLR, 2020.

**Reproducibility:**

4: Could mostly reproduce the results, but there may be some variation because of sample variance or minor variations in their interpretation of the protocol or method.

**Reviewer Confidence:**

4: Quite sure. I tried to check the important points carefully. It's unlikely, though conceivable, that I missed something that should affect my ratings.

---

> ### Author Rebuttal · Authors · 2023-08-28
>
> We appreciate the detailed and valuable comments. We carefully address all your concerns one by one as follows:
> 1. Thank you so much for the valuable and thoughtful suggestions. We will make necessary reorganizations to the relevant part according to your suggestions in the revision.
> 2. The basic breadth aggregation scheme is susceptible to another problem called over-squashing [1], which means that there will be an information overflow for the node as the propagation goes on for many layers. The paper also presents that while some modifications about the breadth aggregation alleviate the over-squashing problem (like attention), they still suffer from that when the distance grows further. Hence, we tried a new approach that utilizes depth aggregation instead of typical breadth aggregation.
> 3. We appreciate the valuable and thoughtful suggestions. We further compare the proposed method to GCNII. The results are shown below:
> | Mean(k=1-5)/(k=6-10) | L=1  | L=2  | L=3  | L=4  | L=5  |
> |---|---|---|---|---|---|
> | GCNII | 89.58/63.57  |  90.59/64.73 | 89.17/64.20  | 91.01/65.35  |  89.87/63.68 |
> | DepWiGNN | 94.21/72.87  | 94.21/72.87  | 94.21/72.87  | 94.21/72.87  | 94.21/72.87  |
>
> L means the number of layers.
>
> The performance of GCNII does not decrease much when the layer number increases, yet it still underperforms our model by a large margin, which demonstrates our model’s capacity to collect long dependency. This may also serve as evidence that depth aggregation might be more effective at dealing with the over-smoothing problem than basic breadth aggregation. We will add this result to our experiment in section 4.4.
>
> Answers to the questions:
> 1. Entity in this paper refers to the spatial roles defined in [2]. For example, in stepgame, “X is to the left of K and is on the same horizontal plane.” the entities will be “X” and “K”; in ReSQ, “A small orange apple is inside and touching box one,” the entities will be “a small orange apple” and “box one.” We will make this point clearer.
>
> 2. For StepGame, all the entities are represented by single capitalized letters from A to Z, so we just recognize them and mark their location during the tokenization process. For SPARTUN and ReSQ, we use nltk RegexpParser (https://www.nltk.org/api/nltk.chunk.regexp.html) with self-designed grammars r""""NP:{<ADJ|NOUN>+<NOUN|NUM>+}""" and r""""NP:{<ADJ|NOUN>+<VERB>+<NOUN|NUM>}""" to recognize entities. We will further elaborate the implementation details to make the process clearer.
>
> 3. The motivation behind treating [CLS] token as an edge feature is that it can help the model better understand the atomic relation (k=1) between two nodes (entities) so as to facilitate later depth aggregation. A straightforward justification can be found in Table 1, where all three PLMs achieve very high accuracy at k=1 cases by using the [CLS] token, which demonstrates that the [CLS] token favors the understanding of the atomic relation.
>
> 4. The most time-consuming part is the shortest path finding between each pair of indirectly connected nodes in LDC, since the edges do not have weights, its complexity is O((n+e)*n), where n and e are the number of nodes and edges, respectively. We will add relevant complexity analysis to the paper.
>
> - [1] Alon, Uri, and Eran Yahav. "On the bottleneck of graph neural networks and its practical implications." arXiv preprint arXiv:2006.05205 (2020).
> - [2] Kordjamshidi, Parisa, Marie-Francine Moens, and Martijn van Otterlo. "Spatial role labeling: Task definition and annotation scheme." Proceedings of the Seventh conference on International Language Resources and Evaluation (LREC'10). European Language Resources Association (ELRA), 2010.

---

### Official Review · Reviewer_jQ72 · 2023-08-04

**Soundness:** 3

**Excitement:**

3: Ambivalent: It has merits (e.g., it reports state-of-the-art results, the idea is nice), but there are key weaknesses (e.g., it describes incremental work), and it can significantly benefit from another round of revision. However, I won't object to accepting it if my co-reviewers champion it.

**Paper Topic And Main Contributions:**

The paper proposes a depth-wise graph neural network for multi-hop spatial reasoning in text. The model can capture long dependency without the need of stacking layers and avoid the issue of over-smoothing. The experiment results demonstrate its effectiveness.

**Reasons To Accept:**

(1) The paper proposes a novel GNN model for spatial reasoning.
(2) The writing is well-organized and easy to follow. Each module in section 3 Method is clear.
(3) The experiments are relatively detailed and the results demonstrate the effectiveness of the proposed model.

**Reasons To Reject:**

(1) The model needs to first find the shortest path between two nodes using a breath-first search. It seems to require a lot of time and space overhead. But the authors do not explore it.
(2) Actually, the main idea of the paper is the depth-wise graph neural network, which is realized by breath-first search and DepWiGNN Module. As there are already lots of relevant studies, the novelty is limited.
(3) The authors argue that the model can avoid the issue of over-smoothing, but there are no experiments to verify it.

**Reproducibility:**

4: Could mostly reproduce the results, but there may be some variation because of sample variance or minor variations in their interpretation of the protocol or method.

**Reviewer Confidence:**

4: Quite sure. I tried to check the important points carefully. It's unlikely, though conceivable, that I missed something that should affect my ratings.

---

> ### Author Rebuttal · Authors · 2023-08-28
>
> We appreciate the thoughtful and valuable comments. We carefully address all your concerns one by one as follows:
> 1. Thanks so much for suggesting the analysis of the algorithm complexity of the proposed method. The time complexity of finding the shortest path between each pair of indirectly connected nodes is O((n+e)*n), as the edges are unweighted. The space consumption is not mainly from the breadth first search but the two-dimensional TPR node memory (as mentioned in the limitation). However, we believe it is worth applying such a node memory scheme as it considerably facilitates the reconstruction of the depth information. We will add the detailed algorithm complexity analysis.
> 2. Sorry about the confusion regarding the main idea of the proposed method. We would like to clarify that the novelty of this paper is NOT about the BFS. It comes from mainly two points: (i) the depth aggregator that aggregates information over depth dimension, and (ii) the node memory scheme that could easily store and reconstruct depth-wise information for each pair of nodes. To the best of our knowledge, most of the research still focuses on the breadth aggregation scheme, some of them concerned with expanding the aggregation target range (as mentioned in the GNNs part of related works), but none of them studied directly aggregating information over depth dimension and using a two-dimension TPR node memory to store and reconstruct depth information between two nodes as we did. We will make this point clearer.
> 3. We, indeed, conduct experiments to verify the over-smoothing issue. Following the standard experimental settings in previous studies [1,2], the experiments in section 4.4 can prove that our model can better handle the over-smoothing problem. In the paper, we compared our model with three different typical GNNs with different layer numbers. The result shows that other models’ performance decrease as their number of layer increases and our model consistently outperform all of them, which demonstrates that our model could better tackle the over-smoothing problem and has better capacity to collect long dependency.
>
> We also followed the suggestion from another reviewer and did one extra experiment for the model GCNII [3], which is a popular model designed to overcome the over-smoothing problem. The results are shown below:
>
>
> |  Mean(k=1-5)/(k=6-10) | L=1  | L=2  | L=3  | L=4  |L=5 |
> |:---:|:---:|:---:|:---:|:---:|:---:|
> | GCNII  | 89.58/63.57  |  90.59/64.73 | 89.17/64.20  | 91.01/65.35  | 89.87/63.68 |
> | DepWiGNN  | 94.21/72.87  | 94.21/72.87  | 94.21/72.87  | 94.21/72.87  | 94.21/72.87 |
>
> L means the number of layers.
> While the performance of GCNII does not decline too much with increasing layer number, it underperforms our model by a large margin. This also proves our model’s immunity to the over-smoothing problem. We will add this result and further highlight the analysis of the over-smoothing issue to better address your concern.
>
> - [1] Li, Qimai, Zhichao Han, and Xiao-Ming Wu. "Deeper insights into graph convolutional networks for semi-supervised learning." Proceedings of the AAAI conference on artificial intelligence. Vol. 32. No. 1. 2018.
> - [2] Wu, Gongce, et al. "QPGCN: Graph convolutional network with a quadratic polynomial filter for overcoming over-smoothing." Applied Intelligence 53.6 (2023): 7216-7231.
> - [3] Chen, Ming, et al. "Simple and deep graph convolutional networks." International conference on machine learning. PMLR, 2020.

---

### Meta-Review · Area_Chair_MtqF · 2023-09-19

**Recommendation:** 3

**Metareview:**

**Summary:** The paper proposes a novel depth-wise graph neural network (DepWiGNN) for multi-hop spatial reasoning in text. The paper claims that DepWiGNN can mitigate the over-smoothing issue faced by GNNs through aggregating information over the depth dimension of the graph. The proposed Transformed Product Rule (TPR) mechanism enables convenient memory updating and retrieval operations through simple arithmetic operations, rather than relying on additional neural layers. Empirical evaluation is performed on two multi-hop spatial reasoning datasets where DepWiGNN outperforms three GNN based baselines.

Overall the paper is well written and easy to follow (with detailed explanation for specific design choices). The empirical results are sound and show that their approach achieves strong gains over multiple strong baselines on two multi-hop spatial reasoning benchmarks. The main concern raised by the reviewers is the limited empirical evidence in the paper pertaining to the main claim of their approach mitigating over-smoothing faced by classical GNNs. Through the rebuttal and reviewer discussion, the authors have added comparison with a stronger over-smoothing mitigating baseline GCNII, as well as expanding the results for the GraphConv model to 10 layers. The results show that the DepWiGNN approach avoids over-smoothing, however the strong empirical performance cannot be solely attributed to mitigating this issue. The paper's empirical claim needs to be strengthened by highlighting that baseline methods suffer from a bottleneck when aggregating longer paths, and how DepWiGNN avoids this.

**Recommendations for Improvement:** (i) Restructure the prose writing to mitigate the concern about the methodology of the paper lacking motivation for each individual design choice of the approach.

(ii) Add details around the time and space overhead complexity of the BFS to the paper.

(iii) Add additional experiments that highlight that baseline methods suffer from a bottleneck when aggregating longer paths, and how DepWiGNN avoids this.

---

### Decision · Program_Chairs · 2023-10-07

**Decision:**

Accept-Findings

**Comment:**

**Summary:** The paper proposes a novel depth-wise graph neural network (DepWiGNN) for multi-hop spatial reasoning in text. The paper claims that DepWiGNN can mitigate the over-smoothing issue faced by GNNs through aggregating information over the depth dimension of the graph. The proposed Transformed Product Rule (TPR) mechanism enables convenient memory updating and retrieval operations through simple arithmetic operations, rather than relying on additional neural layers. Empirical evaluation is performed on two multi-hop spatial reasoning datasets where DepWiGNN outperforms three GNN based baselines.

Overall the paper is well written and easy to follow (with detailed explanation for specific design choices). The empirical results are sound and show that their approach achieves strong gains over multiple strong baselines on two multi-hop spatial reasoning benchmarks. The main concern raised by the reviewers is the limited empirical evidence in the paper pertaining to the main claim of their approach mitigating over-smoothing faced by classical GNNs. Through the rebuttal and reviewer discussion, the authors have added comparison with a stronger over-smoothing mitigating baseline GCNII, as well as expanding the results for the GraphConv model to 10 layers. The results show that the DepWiGNN approach avoids over-smoothing, however the strong empirical performance cannot be solely attributed to mitigating this issue. The paper's empirical claim needs to be strengthened by highlighting that baseline methods suffer from a bottleneck when aggregating longer paths, and how DepWiGNN avoids this.

**Recommendations for Improvement:** (i) Restructure the prose writing to mitigate the concern about the methodology of the paper lacking motivation for each individual design choice of the approach.

(ii) Add details around the time and space overhead complexity of the BFS to the paper.

(iii) Add additional experiments that highlight that baseline methods suffer from a bottleneck when aggregating longer paths, and how DepWiGNN avoids this.